# Overtourism in Iceland: Fantasy or Reality?

**Anna Dóra Sæþórsdóttir [1],\*** , **C. Michael Hall [2,3,4]** and **Margrét Wendt [1]**

[1]  Department of Geography & Tourism, Faculty of Life and Environmental Sciences, University of Iceland, 101 Reykjavík, Iceland; maw6@hi.is

[2]  Department of Management, Marketing and Entrepreneurship, University of Canterbury, Christchurch 8140, New Zealand; michael.hall@canterbury.ac.nz

[3]  Department of Service Management and Service Studies, Lund University, Campus Helsingborg, 25108 Helsingborg, Sweden

[4]  Geography Research Unit, University of Oulu, 90014 Oulu, Finland

\*  Correspondence: annadora@hi.is; Tel.: +354-5254287

**Abstract:** Iceland has been one of the main destinations that have been incorporated into the discourse of overtourism. However, Iceland is different to many other supposed overtourism destinations in that its tourism is based on natural areas. Nevertheless, destination discourses can play an important part in influencing tourist decision-making and government and industry policy making. A media analysis was conducted of 507 online media articles on overtourism in Iceland that were published in 2018, with the main themes being identified via content analysis. The results indicated that the media discourse represented only a partial picture of overtourism and the crowding phenomenon in Iceland, with mechanisms to respond to crowding, the satisfaction level of tourists with their Icelandic nature experience, and local people's support for tourism being underreported. Some of the findings reflect that of other media analyses. However, there are considerable discontinuities between media representations and discourses of overtourism in Iceland, which highlight the importance of national- or destination-level media analysis. The media analysis illustrates the need for a better understanding of different destination discourses and their influence.

**Keywords:** overtourism; crowding; carrying capacity; commons; Iceland; media; discourse; destination change

## 1. Introduction

The concept of overtourism has become increasingly used in media and academic commentaries about the sustainability of popular tourist destinations and their capacity to manage further growth. Overtourism has become the latest term used in the sustainable tourism literature to describe the tensions that exist between tourists, the tourism industry, and permanent residents in the sustainable development of destination communities and the creation of more sustainable forms of tourism [1–3]. Growing numbers of tourists at both natural and urban sites have caused various problems of crowding and business and residential displacement, which impact the quality of life for local inhabitants and the quality of the tourists' experiences and contribute to changes in the nature of the destination [1,2].

Iceland is one of the destinations that have been most associated with the concept of overtourism in the international media discourse on the topic. For example, in an article in *Conde Nast Traveller*, Iceland is described as a "small country on the edge of Europe overrun by masses of selfie-taking tourists" [4], while an article in the *Houston Chronicle* states: "I was about to write Portugal off as another Iceland or Thailand—irreversibly tarnished by overtourism" [5]. A social media article on airfarewatchdog commented: "It's hard to escape that nagging feeling of being just another cog in

Iceland's overtourism and, worse, contributing to the ecological damage of the very landscape you came all that way to admire" [6].

The media discourse over the negative impact of tourism in Iceland reflects the enormous increase of international tourist arrivals in Iceland since the end of the global financial crisis. In 2010, there were about 460,000 international tourists in Iceland; by 2018, that had increased to 2.3 million [7]. Few countries in the world have witnessed as rapid an increase in tourist arrivals since 2010 as Iceland [8]. While the average annual growth in tourist arrivals was 6% to both Europe and the world from 2010-19, the growth in Iceland was 17% [9].

The largest annual increase in international tourist arrivals to Iceland happened in 2016, when growth over the previous year was almost 40% [7]. It is therefore no coincidence that the same year also saw the term "overtourism" being used for the first time in the media to refer to tourism development in Iceland [10]. Prior to the impacts of COVID-19, Iceland was increasingly being featured in media so-called "not-to-go lists", such as on the list compiled by the British newspaper the *Independent* of places not to visit in 2020 [11]. The argument being given was that there were so many visitors at some nature attractions that they had to be closed due to their ecological sensitivity to disturbance [11]. *The Insider* has also put Iceland on a list of 22 places that were "ruined by tourists over the past decade", stating: "It's hard to think of somewhere that's been a more 'it' destination in the past decade than Iceland. There are concerns, however, about the environmental impact of the increase in tourism, impact on quality of life for locals, and whether the tourism boom is a bubble that could burst" [12].

Iceland has a number of environmentally sensitive ecosystems in areas that attract substantial numbers of tourists [13]. Frequent volcanic eruptions and volcanic soils as well as a short growing season means that the environment is vulnerable to external physical impacts, such as tourist visitation [13]. However, the sustainable management of areas with high natural values is made difficult because the natural environment is Iceland's main tourist attraction, with 92% of international visitors claiming nature as the primary reason for their trip to Iceland and perceiving it as "unique" and "unspoilt" [14].

Welcoming two million visitors has proved to be a challenge for Iceland, a country that is the most sparsely populated country in Europe and has only about 350,000 inhabitants [15]. On an annual basis, tourists outnumber locals in Iceland by a ratio as high as 6.5:1, thus putting Iceland in 13th place on a ranking of countries with the highest ratio of tourists per inhabitant [16,17]. As a result, Iceland appears in media discussion as a poster child for what to avoid. For instance, in an article in *Conde Nast Traveller*, a tour leader in the Azores says: "We don't want to be 'the next Iceland.' It's a cautionary tale. You look at the stress that mass tourism has brought on to that country and you say, 'Okay, what can we do to not make those same mistakes?'" [18].

In 2019, the number of international visitors to Iceland declined for the first time in 10 years. Arrivals declined by 14.1% compared to the previous year but this was still higher than in 2016 [7]. While the decline in 2019 was highly appreciated by some, it was a concern to the tourism industry and to government, given the economic and employment significance of the sector. Industry stakeholders are worried about whether the negative publicity has affected Iceland's attractiveness as a tourist destination.

It has long been recognized that the media has a major role in influencing tourist destination perception and demand as well as development trajectories [19] and in influencing decision-making [20]. For example, in democratic states, the media is usually seen as both influencing and being representative of public interest in a topic. Media focus on specific tourism issues has been shown to influence politicians' and decision-makers' attention on environmental concerns and tourism policy as part of an issue-attention cycle by which issues rise and fall on the media and political agenda [20]. As a result, any understanding of sustainable tourism development in a destination ideally requires knowledge not only of the on-the-ground impacts of tourism but also of how the media portrays tourism and its effects in that destination. Therefore, the aim of this paper is to analyze the media discourse on overtourism in Iceland and to discuss how overtourism reveals itself based on existing studies on

crowding and tourism carrying capacity in Iceland. The study is based on a media analysis of online media content, i.e., news articles and blog entries written in 2018 in English, and it seeks to identify the ways in which the international media has presented a narrative of overtourism in Iceland, including how it is explained, manifested, and responded to. The paper, furthermore, analyzes some of the issues underlying overtourism discourses in Iceland as well as some of the means the country has tried in dealing with it. Even though COVID-19 has led to a dramatic drop in the numbers of international arrivals in the short-term, the negative images of overtourism that have seemingly become associated with Iceland still need to be addressed, given Iceland's positioning in terms of its high natural and landscape values [21] and the country's future attractiveness to tourists [22].

The paper is organized as follows. The next section presents the theoretical background, which provides the foundation on which the paper is based. Then the study area and research methods are described, followed by an analysis of the results. The paper concludes by discussing the media presentation and the results from the various studies as well as further considerations and research recommendations.

## 2. Background

### 2.1. The Overtourism Concept and Common Pool Resources

Overtourism refers to a situation where the number of tourists at a destination and the nature of the tourism industry is perceived to be diminishing the quality of life of residents, the quality of experiences of tourists, and the quality of the physical environment, including both cultural and natural heritage [2,23–25]. These fundamental topics have been a major focus of tourism and outdoor recreation research since the 1960s [24], although the terms that describe tourism impacts on destinations have changed over time. As such, overtourism is very closely linked to the concepts of sustainability and tourism carrying capacity [26,27]. The notion of tourism carrying capacity (TCC) was commonly used in the 1980s and 1990s, with the subject matter then becoming transformed into concerns over sustainability, while, more recently, the issue of overcrowding has emerged as a specific aspect of sustainable tourism [21,28]. Nevertheless, all these terms have a core idea, which is to identify when a destination has so many visitors that it is running down natural and human/social capital at a rate faster than it can be renewed; this is also framed with respect to a destination exceeding its carrying capacity and becoming unsustainable [23,27].

The manifestation of overtourism is commonly framed in the form of crowding [29]. In natural areas it can also be seen to result, for example, in damage to vegetation, littering, wildlife and ecosystem disturbance, water and marine pollution, and contributions to climate change [30,31]. In urban settings, negative impacts of large numbers of tourists include increased noise, localized inflation, and a decline in residential housing availability, to name a few [2,23,25]. These negative impacts can, in turn, lead to a decrease in the quality of life of residents and an increasingly negative attitude towards tourists and tourism [32].

Many elements of tourist destinations, such as streetscapes and public and green space, are a common pool resource, while national and regional art galleries, heritage sites, protected areas, and museums are also typically publicly owned [33]. It is usually accommodation, restaurants, and some products, such as casinos, entertainment, and themed attractions, that are privately owned [34,35]. However, the tourist customers of private businesses are also users of shared and public resources and spaces. Accordingly, tourist and tourism-industry utilization of such resources is often regarded by residents to not be in their interest [33]. Overtourism, whether in urban, rural, or natural areas, is therefore related to the use of the "commons" [36].

Overtourism and the concept of the tragedy of the commons are closely related ideas [24,37,38]. The tragedy of the commons is built upon an assumption that there are limits to the extent to which public resources/the environment can be shared and used, while overtourism focuses on overuse of common/shared resources at a destination by tourists and the tourism industry. Both concepts

have evolved from rather rigid, deterministic frameworks to more normative, contextual notions [38]. The mutual characteristics of common pool resources and tourist destinations are:

1.  Ownership of resources is held in common, including via public ownership, or shared by a large number of owners.
2.  Individual users utilize the resource for personal benefit. It is often in the interest of commercial users to utilize the resource as much as possible to obtain additional revenue. However, the loss due to overuse, which may be a financial loss, a reduction in personal satisfaction level, or a reduction in access, is shared among all users. This can lead to overuse of the resource.
3.  No private individual is usually willing to invest with the aim of improving the resource as there is no guarantee that the return to investment would go back to the private investor. This is why government is usually the institution responsible for improvements, either via direct investment or regulation.
4.  Control of access to the resource is difficult. This can be for a variety of reasons. For example, boundaries may be difficult to delineate and police, the size or area of the resource may be very large, or control may not be accepted due to political reasons, including that it is a common and/or public space.

By their very nature, tourists and the tourism industry "consume" tourist destinations by utilizing, to a greater or lesser extent, its various tangible (e.g., landscape, parks, and green space) and intangible (e.g., culture, atmosphere) common and public resources. The development and use of some of the resources is especially planned for tourism and promoted to tourists by, for example, visitor centers and destination marketing organizations, but often tourists become unplanned users of local resources, for example through the discovery of attractive viewpoints [39]. The development of tourism at a destination invariably means that the appearance of a place changes. Nature destinations become less natural and elements of the rural landscape become commoditized and change, as does the townscape. Thus, tourism facilities become embedded within regions but may also sometimes stand out as isolated elements in space [34]. These elements of tourism development can also, paradoxically, sometimes change or diminish the overall attractiveness of the destination, depending on what the attraction is [40,41]. Tourism can therefore have very real effects with respect to the consumption of environments, landscapes, and places [39].

### 2.2. Managing Overtourism

One of the earliest ideas for managing the "overconsumption of areas" was to set use limits or "caps" on the number of visitors at a destination [39]. The idea was introduced by the US national park service [42] due to crowding and is the core of the idea of tourism carrying capacity (TCC). TCC has, since the 1960s, been used in wilderness and tourist destination management. It assesses the impacts of tourism from the point of view of the impact of visitors on the environment and the physical carrying capacity as well as from the point of view of the impact people have on other people, that is, the psychological carrying capacity [43,44]. Carrying capacity is also at the heart of Butler's tourism area lifecycle (TALC) model, where he argues that, when the TCC of a destination is reached, the destination will potentially decline and lose its attractiveness or, in other words, become the victim of overtourism and become unsustainable [40].

The perception of crowding occurs when the sociocultural carrying capacity is overstepped, a tipping point usually defined by personal and community norms [45–50]. Norms refer to criteria that are used to evaluate behavior and the environmental and sociocultural conditions of the destination [51]. Norms can be differentiated as either social or personal norms. Personal norms refer to the expectations of an individual, e.g., an expectation that a nature experience would be characterized by solitude, making the individual more sensitive to crowding [45,52,53]. In contrast, social norms are shared by different members of a group, but they can also vary between nationalities and social groups [45,54]. Social norm theory assumes that there is a group agreement or consensus about suitable social and

environmental conditions at a destination, which can be used to create standards of quality based on users' preferences [45,48,54]. The normative approach has mainly focused on issues of crowding in terms of encounter norms. These define the number of other people a person can tolerate meeting or having contact with at a destination within a given time [51].

If visitors have normative standards regarding the various aspects of their experiences, then such norms can be used to help set basic standards of quality to maintain or aim for [51]. By doing so, social carrying capacity estimates can be set and management actions undertaken [48] to satisfy the majority of site visitors. In other words, overtourism can essentially be avoided. The options of evaluating limits or setting caps have mostly been limited to single tourist sites rather than entire destinations [28]. An explanation for this lies in the growth model that underlies most tourism [55,56]. Since the 1960s and the development of mass tourism, an increase in tourist arrivals has been regarded as a primary goal in tourism development. The main reason for this is that an increase in tourist arrivals arguably creates various economic benefits, among others, in the form of increased national or regional economic growth and employment options. The general assumption is that limiting tourist numbers would hinder the economic potential of the tourism industry [1,24,26,28,37,57]. This perspective is also represented by several supranational organizations, such as the UN World Tourism Organization (UNWTO), the World Economic Forum (WEF), and the World Travel and Tourism Council (WTTC), all of which advocate for tourism growth [28,58]. Growth is not seen as the root of the overtourism problem but rather ineffective management [26]. For example, this is reflected in the title of the World Travel Market Minister's Summit, coorganized by UNWTO in London in November 2017: "Overtourism: growth is not the enemy, it is how we manage it" [59].

Despite the importance of resident perceptions of sustainability in a tourism context [60], there appears to be only limited awareness among the public about the impacts of tourism and most people are resistant to making significant changes in their travel behavior. Furthermore, the public seem to depend on the government to tackle the problem [61]. Such a situation perhaps reflects the commons problem of people not recognizing their individual contributions to the problem of overtourism and the overall sustainability of tourism [33,41]. Nevertheless, the public and, in particular, the residents of destinations associated with overtourism have increasingly become involved in the discussion about how to manage overtourism and its effects.

Social representation theory and social exchange theory have been used to understand the nature of residents' attitudes towards tourism and societal conflicts due to tourism. Social exchange theory has focused attention on residents' perceptions of the relative economic, environmental, and sociocultural costs and benefits of tourism in their community and their evaluation/satisfaction [62,63]. Communities that receive greater economic benefits from tourism are generally more positive towards tourism. Thus, net economic gain can often be a good predictor of positive attitudes towards tourism [64]. Social representation theory focuses on residents' experiences and beliefs held about tourism and how these are socially constructed in terms of, for example, the media and the image it conveys of tourism [65]. Thus, social representations are mental constructs that help define an individual's reality. According to Moscovici [66], there are three types of social representation. Firstly, there are hegemonic representations that are encouraged by those in control in the society and are often generally believed or approved. Secondly, there are emancipated social representations that are collective within subgroups but are not generally approved. Thirdly, there are polemic representations that result from intergroup conflict and often denote different views or beliefs about a topic. Social representations are organizing principles of symbolic relationships between individuals and groups that various members of a population share common views about [67], e.g., a given social issue, such as the effects of tourism. However, social representation theory implies that variations in these meanings depend upon group memberships held by individuals, as they are anchored in other collective symbolic realities. In certain destinations, mainly European cities, public movements of opposition to overtourism have been formed [1,58,68] and, even though tourism is a major contributor to the Icelandic economy, concerns have been expressed about the extent of tourism in the country [69]. In fact, such opposition

is potentially extremely influential on destination image as the emergence of the term overtourism is rooted in the media coverage of anti-tourism movements in Europe [23,25], although Iceland has not had such a social movement. Therefore, a greater understanding of media discourse and representations of overtourism would seem to be of great significance for sustainable destination management.

### 2.3. Media Discourses and Destination Change in the Era of Overtourism

It has long been acknowledged that change is an inherent characteristic of tourism destinations. "Tourist areas are dynamic . . . they evolve and change over time" [40] (p. 5). However, trying to make sense of the transformations of destinations is challenging because they are not only transformed by local processes but are also heavily influenced by global capital, discourses, and movement [57]. Destinations are constituted at different scales, e.g., individual countries, municipalities, cities, towns, regions, and even tourist resorts. Nevertheless, regardless of the spatial scale, tourist destinations are socially and historically produced spaces that have meanings and identities ascribed to them [57,70,71]. However, as a result of their mutual influence on each other, destination identity and the types of tourists that visit a destination change over time [40,57].

Discourses lie at the heart of tourist destination change. Saarinen argues that "tourist destinations are seen as dynamic, historical units with specific identities characterized by hegemonic and other discourses, which all produce a notion of what the destination is and represents at the time" [57] (p. 161). At any given time, a destination's identity is being influenced by multiple, sometimes even contradictory, discourses. Saarinen, for example, presents two types of discourses that interact to help create the identity of a destination: the discourse of region and the discourse of development [57]. He defines the discourse of region as the idea of the destination—the knowledge and meanings that individuals have acquired through various channels, such as literature, maps, advertisements, television, and social and general media. Discourse of development, however, refers to the material and economic characteristics of a destination, including the number of (international) tourists, the provided infrastructure and services, and policies and strategies. Saarinen [57] argues that, while one discourse is usually more dominant, others still exist and thus there may be competing identities and representations of a destination. These competing identities can cause conflicts with regards to tourism development, for instance if the tourism industry advocate for a different place identity or conception of place than that which residents, or even other industries, can share. This reflects what has happened with respect to the different discourses of sustainable development used by the energy and tourism industries in the Icelandic Highlands [69].

One example of a media discourse that has the ability to impact a destination's identity is the discourse of overtourism. Koens et al. note that "the concept of overtourism has come to prominence as one of the most discussed issues with regards to tourism in popular media" [25] (p. 1). The media plays an important role in influencing tourism demand [19,20] as well as tourism policy and decision-making [20]. The media's power lies in, among other things, its ability to select which events and issues are given attention and which ones are neglected as part of the ecology of news. Moreover, the media can make news by creating stories. The amount and type of coverage that an issue receives also influences how the public perceives the issue [20,29], and the degree of attention given to an issue can rise and fall over time as part of what has been termed an issue-attention cycle [72,73], with consequent implications for decision-making and, in the case of tourism, destination perception. Importantly, both conventional and online media have been found to contribute to agenda setting as a result of attention being given to particular issues, including in relation to the environment [74–76]. However, media coverage does not necessarily reflect the true character and importance of an issue [77–80]. Given the recent media attention that overtourism has received and continues to be given in comparisons with the impacts of COVID-19 on tourism destinations [81,82], it is therefore of importance to improve understanding of the extent to which the media is reflecting actual or exaggerated situations.

Despite the growing use of the overtourism term in media reporting and commentary, there is surprisingly little media analysis of the subject in a tourism context. Phi conducted a content analysis on news articles in English with the aim of shedding light on how the media frames modern overtourism [29]. The social representation framework gives direct attention to systems of benefits, values, attributes, and explanations that individuals hold about tourism [83]. Phi's findings pointed to four themes that the media utilized to represent overtourism: tourists, locals, cities, and the tourism industry [29]. The first theme centered around the causes of overtourism and statistics on the increase in tourist arrivals. The second theme was reports about impacts of overtourism as experienced by local residents. The third theme was that the media paid special attention to overtourism in cities, as opposed to overtourism in national parks and protected areas. The fourth theme emerged around the representation of the tourism industry. The media commonly displayed support for the continued growth of tourism due to the industry's role in creating jobs and revenues for the local communities. Thus, the media did not blame the high number of tourists per se, i.e., overtourism, for negative impacts caused but represented tourism management as the root of the emerging problems. Phi concluded that the media was displaying overtourism in an overly simplistic way as it failed to represent the diverse sides of the issue [29]. In essence, "the current media-led exposé of overtourism is characterized by outrage, sensationalism and hysteria" [1] (p. 4). The media narrative is therefore potentially simplifying the issues of overtourism and fails to present a more considered account of the drivers of overtourism sentiments. Nevertheless, such representations can still influence perceptions of tourism destinations.

## 3. Study Area

Being an island in the North Atlantic Ocean has meant that, historically, Iceland was quite isolated, with transportation limited to irregular boat trips between the island and Europe. That changed completely in 1945 when international passenger flights began and tourism started to expand [84]. Since 1950, the average annual growth in visitor arrivals has been about 10% but, after 2010, a period of dramatic annual average increase of around 22% began, so that, by 2018, tourist arrivals had reached 2.3 million. In 2019, for the first time since the global financial crisis, tourist arrivals declined, down 14.1% from the record 2018 figures [7] (Figure 1). In addition, cruise ship passengers also increased very sharply from 100,000 passengers in 2015 to more than 180,000 in 2019, a 22% average annual increase [7].

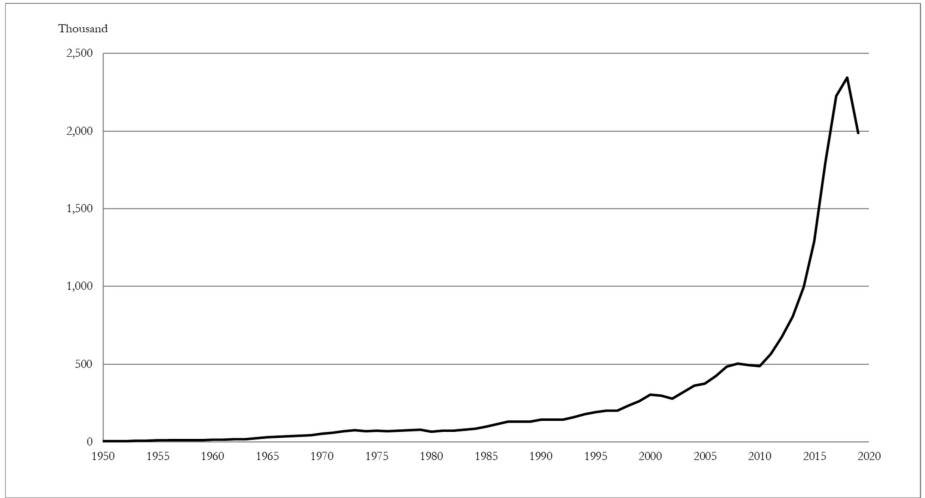

**Figure 1.** The number of international visitors to Iceland. Data are derived from [7].

The rise of tourism in Iceland began shortly after the international financial crisis in 2008 [85]. The crisis led to a drastic decrease in the value of Iceland's local currency (the Icelandic Krona) and thus traveling to Iceland became more affordable [86]. Two years later, the Eyjafjallajökull volcano

erupted and caused a massive disruption of air traffic throughout Europe. As a result, Iceland became the topic of an international discussion, which was further supported by a promotional campaign under the name "Inspired by Iceland". The campaign was facilitated and supported by stakeholders from government and the tourism industry with the aim of marketing Iceland as a safe and attractive destination for tourists [87].

To complicate the annual increase in visitation, tourism in Iceland has for a long time been very seasonal and concentrated in time, with the majority of tourists coming in the summer. After the rapid increase in visitor arrivals, seasonality has been reduced in relative terms, but this development is, so far, spatially limited mainly to the capital area and the south coast [21]. In 2010, half of all overnights in the capital area were in May–August, while, in other regions, 83% of all overnights were in that period. In 2019, this ratio had changed to 36% in the capital and 58% in other regions [88]. Occupancy rates of hotels increased from 46.9% in 2010 to 64.5% in 2019 [89]. The average length of stay was 6.6 nights in 2019 [90].

The sharp increase in international visitor arrivals has had numerous impacts on Iceland's economy, environment, and society [91]. The tourism industry played a vital role in helping Iceland recover from the financial crisis in 2008 [85] and, by 2017, had become, by far, the largest export sector in Iceland, with 42.0% of foreign exchange earnings, compared to 26.4% in 2013 [92]. In 2010, tourism activities were associated with 3.4% of the gross domestic product (GDP) and, in 2018, this had increased to 8.1% [93]. In addition, the supply for all kinds of services has increased, including restaurants and retail, both in Reykjavík and in rural areas. The tourism industry had also become the main provider of new jobs after the financial crisis [69]. Thus, most Icelanders are generally positive towards tourism, although somewhat less so in recent years, with those who are positive towards tourism decreasing from 80% in 2015 to 68% in 2018 [94]. However, tourist destinations in Iceland suffer from very uneven geographical distribution of visitation in space and time. Studies from the Icelandic Tourist Board show that some destinations are heavily visited, such as the capital Reykjavík, which was visited by 92% of international visitors, Geysir (82%), Þingvellir (70%), and Mývatn (73%) [7,14] (Figure 2). During summer, the attitudes of residents are more negative, as about 27–30% state that the number of tourists was rather high or too high, compared to only 3–7% in winter [95,96]. Crowding is also a bigger problem in the most visited regions in the country. About 77% of residents at Lake Mývatn, 42% in South Iceland, and 30% in the capital area think that there are somewhat too many tourists or way too many [97].

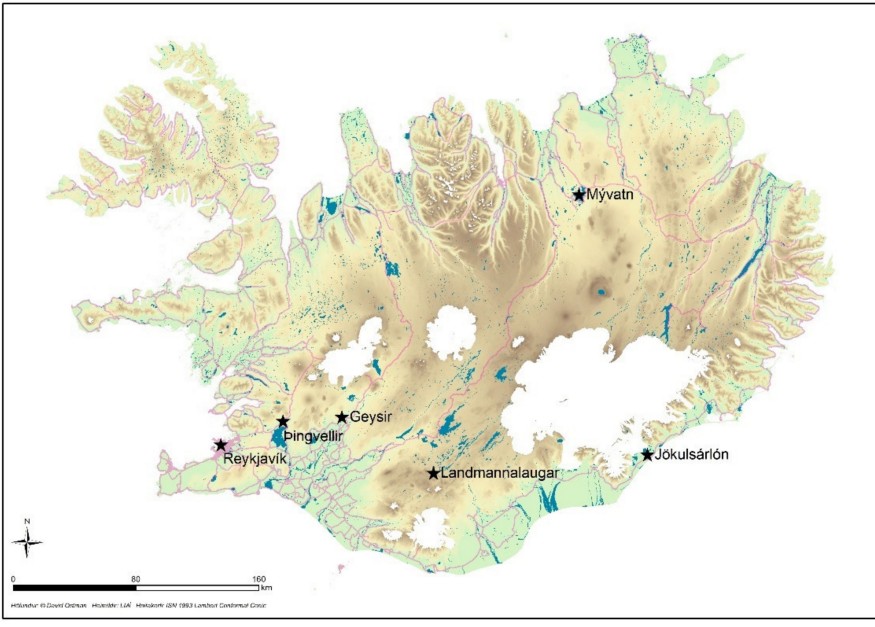

**Figure 2.** Some of the popular tourist destinations in Iceland.

The Icelandic environment is highly susceptible to human impact. Thus, environmental degradation and the poor condition of hiking trails due to visitor pressure is evident [13,98]. At Þingvellir National Park, Geysir, and Jökulsárlón, over one-third of visitors complain about there being too many visitors. Although there is a substantial variation between summer and winter, with winter being perceived as far less crowded [21,99,100]. The main highland destination of Landmannalaugar is also perceived as crowded [101,102]. Perceptions of overcrowding also exist in urban areas, with one-third of summer visitors and 16% of winter visitors perceiving Reykjavík as having too many visitors [99,100]. Nevertheless, it is important to emphasize that tourists continue to be very satisfied with their overall experience in Iceland and at the individual destinations, despite crowding at certain locations [14,21].

## 4. Methods

In order to explore the Icelandic overtourism narrative, this paper reports on a content analysis of media that report overtourism in Iceland. Content analysis is a well-established research method for making replicable and valid interferences from media and communication data, including in relation to tourism [103,104]. Content analysis involves the "careful, detailed, systematic examination and interpretation of a particular body of material in an effort to identify patterns, themes, biases and meanings" [87] (p. 338) and is often used as an analytical framework for comparing media content to "real world" situations [92]. Quantitative content analysis, as primarily used in the present study, identifies the frequency with which key words and concepts appear in the media under study. However, the identification of conceptual categories was undertaken via a thematic analysis of the media, which led to the development of emergent codes [105]. Analytical reliability was assured via the development of a coding scheme for conceptual variables and subsequent cross-checking between the researchers to help ensure intercoder reliability [103,106].

Every day, the media covers various topics and is available globally on the web, in print, and through broadcasting. Since a full examination of available media content is beyond the scope of this study, it was decided to analyze written online media content, i.e., news articles and blog entries, about overtourism in Iceland written in 2018 in English. Other forms of media (e.g., social media, television, and radio) were excluded from this study. The reason for choosing written content from 2018 is that this is the most recent year during which tourist arrivals continued to grow before declining in 2019. Media content in English was chosen as it can be understood by large numbers of people, given that it is the world's major second language. In addition, English is the mother tongue of two of the largest groups of tourists traveling to Iceland: tourists from the Unites States of America and the United Kingdom [14].

The search for content was organized in cooperation with Promote Iceland, a public-private organization with the aim of promoting and marketing Iceland in foreign markets. It was conducted by using the program Cision, which is used by Promote Iceland to monitor foreign media commentary about Iceland and monitors more than 7 million websites. The program provides access to a far greater number of articles than previous overtourism content analysis studies [29]. Based on the keywords "Iceland", "tourism", "tourists", "visitors", "overtourism", "overcrowding", and their derivatives, e.g., "Icelandic", and various alternative forms of spelling, e.g., "over tourism" or "over-crowding", the program identified a total of 878 online news articles and blog entries from 2018. The articles were then examined with regards to their relevance. Articles that discussed overtourism with no Iceland context as well as articles that discussed tourism in Iceland without an overtourism or overcrowding context were excluded from the analysis along with articles that were no longer accessible online. In the end, 507 articles were identified as relevant for the analysis. Of these, 170 were "original publications" of the articles and the remaining were "reposts". Reposts are here defined as articles that are identical to original articles and republished on at least one other online media website. The extent to which publications are reposted is itself an indicator of the potential power of media stories to influence destination discourses. However, how often the articles were replicated on social media was not analyzed and is beyond the scope of this study. Both original publications and reposts

were analyzed collectively so as to identify the strength of various themes. However, reposts are also identified separately. The articles appeared on 130 different media sites, most of which (72%) were online magazines or news sites, followed by blogs (23%) and company websites for commercial purposes (5%). They were published in 20 different countries, with the majority (28%) published in the USA. Around 26% of the articles were published by international media (i.e., media platforms not associated with a single country) and 16% were published in the UK. Interestingly, in terms of when the articles were published, the majority were produced after the summer, which is the peak period for visitation to Iceland (Figure 3). What is more remarkable in the analysis of the articles examined in this study is that the number of original articles is reasonably constant throughout the year regardless of international arrival levels, but the number of reposts is far greater in the autumn/fall after the period of peak visitation.

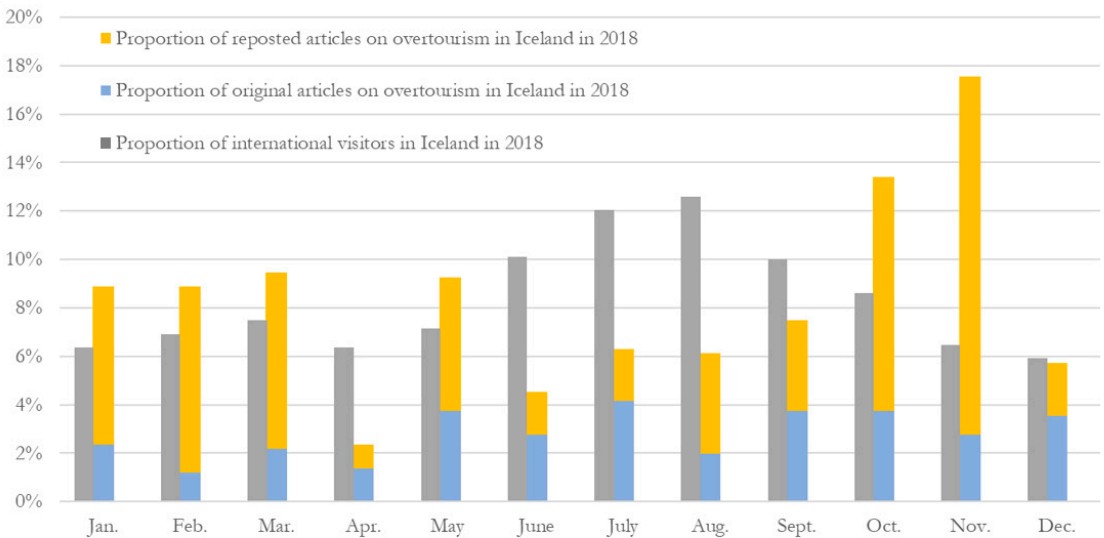

**Figure 3.** The proportion of articles on overtourism in Iceland and the proportion of international visitors in Iceland in 2018 by months.

For this study, the whole text, including the headline, was defined as the unit of analysis. If articles included pictures, these were also taken into consideration. In order to establish content categories, emergent coding was applied [103,105], through which four categories were identified: the extent to which the media portrayed overtourism as problematic in Iceland, overtourism's manifestation in Iceland, the reasons for it, and the reactions to it. Qualitative analysis through open coding assessed the meaning behind themes within each category [107], as presented in the next section. A coding sheet was developed where each article, including its pictures, was classified according to the content categories. All headlines as well as pictures were consistent with the article itself, meaning that they were classified together with the whole text. Finally, descriptive statistics were used in order to organize and quantify the emerging themes and the results were arranged in figures. By that means, regularity and patterns in the data could be highlighted [103].

It is important to note that the results of a content analysis are limited to the specific definitions and category systems of each study. Therefore, care should be taken when comparing the findings of this study with other content analysis studies. In addition, content analysis alone cannot serve as a foundation for broad claims about the impact of media content and it is consequently beyond the scope of this study to assess the direct impact of the discussed articles on their audiences. A limitation, which applies specifically to online content analysis, is also the fact that online content may disappear or become inaccessible [103].

## 5. Results

### *5.1. Media Results*

#### 5.1.1. The Media Portrayal of Tourism in Iceland as Overtourism

The 507 articles that discussed overtourism in relation to Iceland were analyzed with regards to the overarching narrative that they presented. Each article was examined based on its key theme, i.e., in what context it discussed overtourism and Iceland. It was found that the great majority of articles (74.5%) presented that Iceland was suffering from overtourism and that the (whole) country was a "poster child" for overtourism (Figure 4). The degree to which overtourism was problematized varied. While many articles simply named Iceland as a destination suffering from overtourism without further explanation or discussion [108–110], others described how Iceland was "being loved to death" [111,112]. Another common trope was about how tourism had once saved Iceland's economy but was now a threat to the country's authenticity. Other authors claimed overtourism meant that Iceland was at risk of "selling its soul" [112] and was on its way to becoming a "Disneyland" [113]. In addition, it was stated that other destinations feared that they would become "the next Iceland" [18].

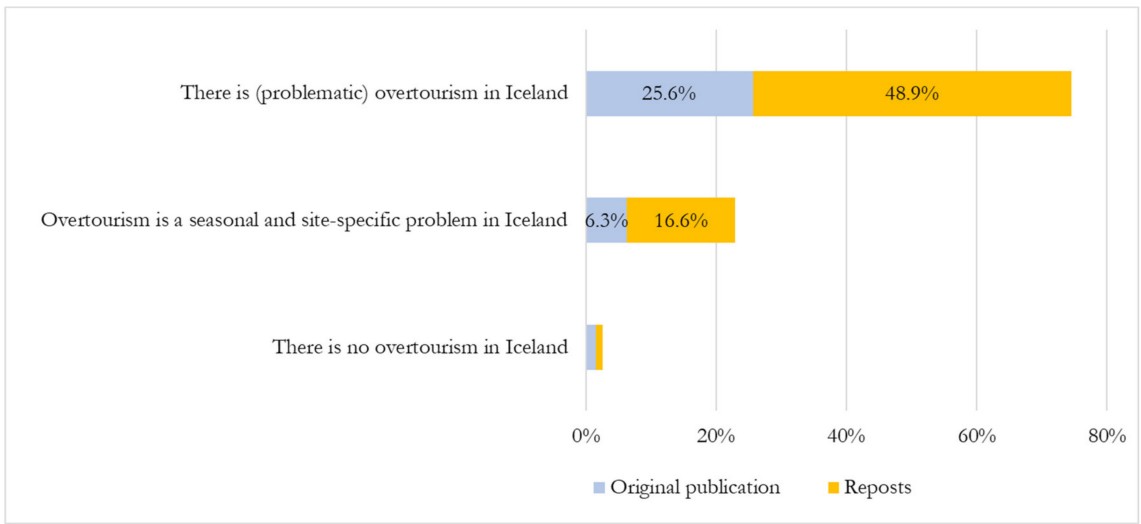

**Figure 4.** The key themes of the analyzed media articles (N = 507).

A closer analysis of the articles highlights that the majority of the articles are reposts, with 337 reposts in all (Table 1). Some of the original sources of reposts include CNN, *The Gazette*/*The Washington Post*, the Australian Fairfax group of newspapers (*Traveller*, *The Age*, *Sydney Morning Herald*, and *Canberra Times*), *The Leader News*/*Global News* (Canada), Agence France-Presse (AFP), and the European Travel Commission. Significantly, in terms of reports, many of the sources are news agencies that syndicate material as well as being publishers in their own right. This, in great part, helps explain much of the reposting of Icelandic overtourism material.

In 22.9% of the articles, the overtourism narrative was criticized based on the argument that it was not a problem for all of Iceland but, instead, it was discussed as a seasonal and/or site-specific problem. The articles mentioned that overtourism in Iceland was mostly limited to Reykjavík, the Golden Circle area, and a few other specific sites in South Iceland, in particular during the summer months. Finally, there were also 13 articles, or less than 3%, that represented the opinion that there was no overtourism in Iceland at all. Of these, the authors of three articles argued that Iceland was a good destination to escape overtourism. The other 10 articles referred to interviews with Icelandic stakeholders who claimed that Iceland was not suffering from overtourism.

**Table 1.** Reposted articles by month of publication, title, and source.

| Month | Title of Article | Appeared On | Number of Reposts |
|---|---|---|---|
| Jan. | Travel tips: Ten things you need to know about travel in 2018 | Traveller/The Age/Sydney Morning Herald/Canberra Times | 9 |
| | 10 Trends That Will Make 2018 More Adventurous | Travel pulse (Canada) | 3 |
| | Iceland is hot, the U.S. is not—and other travel trends for 2018 | Market Watch | 5 |
| | 2018 Travel Hotlist: 20 hot trends and tickets for the year ahead | Independent (Ireland) | 2 |
| | Off-limits no more: Why you should visit these countries with a bad reputation | Traveller/The Age/Sydney Morning Herald/Canberra Times | 13 |
| | The best place to visit every month this year, according to a popular online travel agency | Business Insider | 1 |
| Feb. | Barcelona and Spain and Their Battle With Over-Tourism//Smart Destinations to tackle overtourism | Euromonitor Internationally - Blog | 1 |
| | European Travel Commission Reports Extraordinary Results for European Tourism in 2017 | European Travel Commission | 38 |
| Mar. | Europe's beauty spots plot escape from the too-many-tourists trap | The Guardian | 1 |
| | Overtourism and the big chill: travel trends in 2018 | AFP/Breitbart/France24 | 33 |
| | Avoid the crowds: 7 popular destinations and the places to go instead | The National | 1 |
| | Breaking Travel News investigates: Tourism in Malta | Breaking Travel News | 3 |
| Apr. | Dear dictionaries, this is why 'overtourism' should be your 2018 word of the year | The Telegraph | 5 |
| May | Venice Tourism Checkpoints are a sign of Europe's fractures approach . . . | Skift | 1 |
| | The 9 most at-risk tourist hotspots and where to go instead | Matador network | 1 |
| | Lonely Planet's Top European Destinations Of 2018 Take Aim At Overtourism | Huffpost (Lonely Planet) | 7 |
| | Countries with the most tourists per head of population: Destinations suffering 'overtourism' | Traveller | 2 |
| | Reykjavik Rises, Barcelona Slips And Nice Rebounds As Top European Summer Travel Destinations In 2018 | Global Travel Media/Travel Weekly/PR Newswire | 17 |
| | Holidays 2018: Which country in the world has the biggest population of tourists? | Express | 1 |
| June | Iceland to implement measures to curb over tourism | Travel and Tour World | 2 |
| | Places to avoid and where you should go instead | News.com.au | 2 |
| | Video: Iceland Denies Overtourism as Europe Searches for Solutions | Skift | 2 |
| | GetYourGuide's Branded Tours and 7 Other Tourism Trends This Week | Skift | 1 |
| July | Increased Tourism: The Pressure of Unrestrained Growth | Hua Hin Today | 1 |
| | Cities suffering from overtourism: How to visit as a responsible tourist | Traveller | 1 |
| | Avoid Crowds at Popular Destinations and Try These Hidden Gems | The National Geographic | 2 |
| | Why Norwegian Air Doesn't Worry About Overtourism When It Chooses New Routes | Skift | 1 |
| | Europe Made Billions from Tourists. Now It's Turning Them Away | For immediate release | 3 |
| | EIC lays out tourist roadmap—Measures to control overcrowding could improve the visitor experience | Bangkok Post | 2 |
| | Skift Basics: 10 Essential Reads to Understand Global Travel | Skift | 1 |

**Table 1.** *Cont.*

| Month | Title of Article | Appeared On | Number of Reposts |
|---|---|---|---|
| Aug. | Wish you weren't here: how the tourist boom—and selfies—are threatening Britain's beauty spots | The Guardian | 6 |
| | From "Game of Thrones" to "Mamma Mia," set-jetting tourism is everywhere | Quartz | 2 |
| | The other TV and film locations struggling like Cornwall with the Poldark effect | Cornwall Live | 14 |
| Sept. | Overtourism: How you can help solve this worldwide problem. | Nomadic Matt's Travel Site | 3 |
| | Hotels springing up in New Zealand to close room shortage | Travel Weekly | 2 |
| | Experts to Discuss Overtourism in Nation's Capital | Travel Pulse | 3 |
| | Overtourism—The Herman Trend Alert | Expert Click | 1 |
| | 'Game Of Thrones' Tourism Is Overwhelming This Idyllic Croatian Town | Uproxx | 2 |
| | Europe's fastest growing tourist economies include Moldova, Bosnia, and . . . | Quartz | 1 |
| | Hotspots - Where to next for future tourism destinations? | New Zealand Herald | 2 |
| | The View From Europe: Who owns Tourism? | Carribean News/Barbados Advocate | 11 |
| Oct. | Can the world be saved from overtourism? | CNN | 28 |
| | Nature Works: New Research Shows The Economic Value Of National Parks | The Reykjavík Grape Wine | 1 |
| | Iceland visits to learn tourism lessons from New Zealand | Radio New Zealand | 1 |
| | The European Island Paradise That American Tourists Have Yet to Discover | Press from | 2 |
| | Ethical tourism: Steps to cut the negative effects of travel | The West Australian | 2 |
| | Overtourism: Time for some limits | The Garden Island | 1 |
| | How the Azores Will Hold Off the Crowds and Stay a Natural Wonder | Conde Nast Traveler | 1 |
| | Iceland Is Not Overrun With Tourists, Despite What Everyone Says | Conde Nast Traveler | 5 |
| | The Backlash againt overtourism | The Economist | 2 |
| Nov. | WOW! Icelandair buys a competitor | Houston Chronicle/The Telegraph | 5 |
| | Detouring: Top world destinations are overrun. Take these suggestions for . . . | The Gazette/The Washington Post | 24 |
| | Overbooked and overlooked—travel destinations to visit in 2019 | IOL News/Safrica24 | 2 |
| | Hanoi's 'train street' becomes selfie central | CNN | 26 |
| | How Instagram has transformed how people choose their next vacation destination | The Leader News/Global News | 21 |
| Dec. | More people will be visiting Portugal in 2019—here's why | Conde Nast Traveller | 1 |
| | Hidden Iceland: Avoiding the Crowds | Forbes | 1 |
| | Most popular destinations for Australians in 2018: Top 10 places on Traveller | Traveller | 1 |
| | Big Sur plea: Tourists, honor our home—Overwhelmed by crowds, locals launch 'Big Sur Pledge" campaign for better behavior | The Mercury News | 4 |
| **TOTAL** | | - | **337** |

## 5.1.2. The Manifestations of Overtourism in Iceland as Presented by the Media

The second category of themes that emerged from the media analysis was how overtourism was manifested in Iceland. Overall, there were 158 articles (31.6% of the total number of articles)

that did not specify in what ways overtourism manifests itself in Iceland. These articles argued that overtourism and overcrowding were an Icelandic reality without discussing it further. The remaining articles were analyzed with respect to what the authors referred to in order to paint a picture of overtourism in Iceland. Some articles referred to multiple ways in which overtourism appears in Iceland, whereas others only mentioned one aspect.

The most common way used by articles to paint a picture of overtourism in Iceland was numerical references to the industry's growth as well as references to the population size of Iceland; this was noted in 51.3% of the articles (Figure 5). Unhappy locals and a decrease in their quality of life was the second most frequent manifestation, mentioned in 20.7% of the articles. It was argued that locals experienced a disruption of their daily lives due to tourism. An increase in pricing, in particular for housing, was commonly named as a negative impact of overtourism. Third, 16.7% of the articles described how high visitation was having a negative impact on natural and/or fragile tourist sites in Iceland and thus referred to a decline in their quality. The fact that certain sites had been closed off for tourists was often considered a key manifestation of the overtourism problem. Fourth, 5% of the articles presented various negative tourist behaviors, of which human waste and littering were the most frequently mentioned. Seven articles (1%) also presented that tourism was impacting the physical appearance not only of nature sites but also of urban spaces. They mentioned that Reykjavík in particular was being transformed as more hotels were being built and nonlocal stores, such as Dunkin Donuts, appeared. Finally, there were two mentions (0.4%) relating to the reliance on migrant workers to meet the growing tourist demand as well as one mention (0.2%) of the argument that signs, bills, and other written material were increasingly written in English rather than Icelandic.

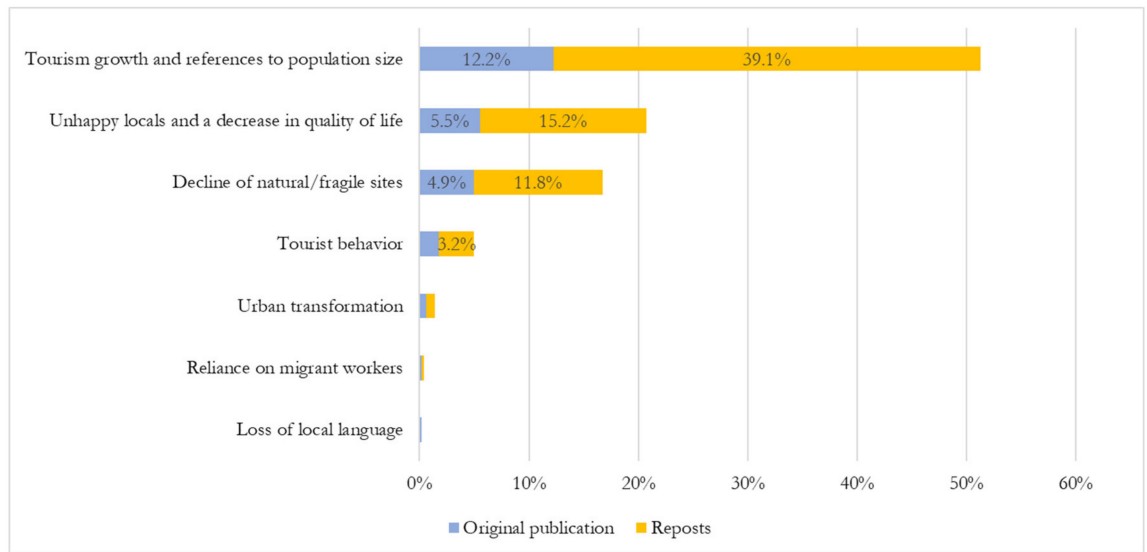

**Figure 5.** Manifestation of overtourism in Iceland as presented by the media (N = 507).

## 5.1.3. The Reasons for Overtourism in Iceland

A total of 254 articles (50.1% of the total number of articles) made an attempt to explain why there is overtourism in Iceland. By far the most frequently mentioned cause, mentioned in 30% of all articles, was Iceland's presence on social media (Figure 6). Instagram, which went live in 2010, the same year as the tourism boom in Iceland started, was commonly named as one example. The writers argue that Instagram and its function of "geotagging" contributed to the Icelandic tourism boom and ultimately to overtourism. Television shows and films are also assumed to play a significant role in creating tourism demand and many articles discussed specifically how *Game of Thrones* had increased tourist arrivals to Iceland. *Game of Thrones* was not only said to have created overtourism problems in Iceland but also in Croatia's capital Dubrovnik. Moreover, there were also articles that mentioned influencers, such as Justin Bieber, as social media contributors to overtourism in Iceland.

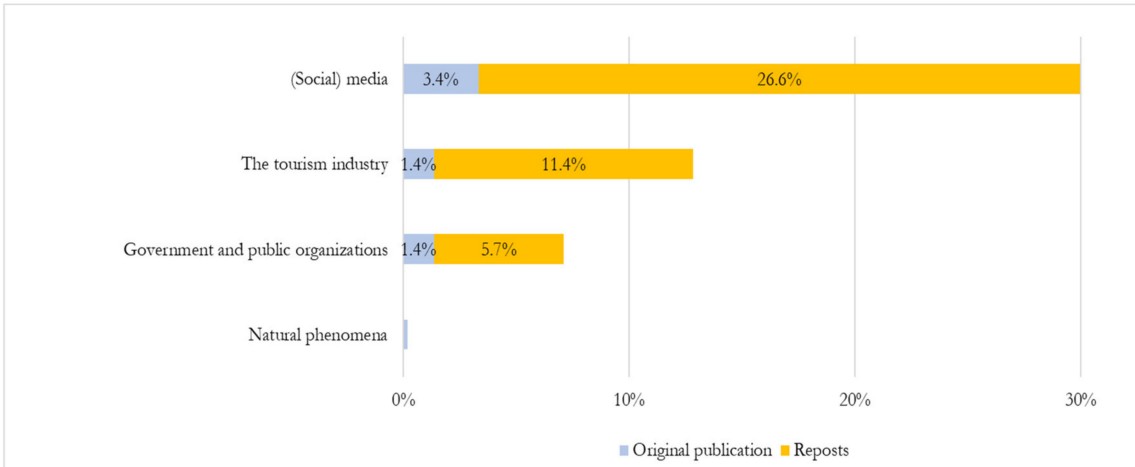

**Figure 6.** Reasons for overtourism in Iceland as presented by the media (N = 507).

The second set of reasons for overtourism in Iceland is related to the tourism industry, which 12.8% of all articles discussed. A few articles mentioned that an increased supply of often cheap flights to Iceland had resulted in uncontrolled tourism growth. However, most mentions were related to the Stopover program offered by Icelandair. The program was said to have helped increase tourist arrivals to Iceland but, since the stopover tourists tend to stay for a short time in Iceland, their visits are often limited to Reykjavík and its surroundings. They thus contribute to overcrowding rather than an even distribution across the whole country.

7% of articles also argued that the government and official organizations in Iceland had facilitated the overtourism problem, in particular through a lack of management of infrastructure, monitoring, and custody as well as through marketing efforts. Finally, there was one article that mentioned that the Eyjafjallajökull volcano in 2010 had contributed to overtourism.

### 5.1.4. Reactions to Overtourism

The articles not only discussed in what ways overtourism was problematic for Iceland but also how the government and official tourism organizations, industry, and tourists have and/or should respond to overtourism. The majority dealt with the reactions of Iceland's government and official organizations (Figure 7). Of these, most articles described how the Icelandic government had attempted to limit (over)tourism, for example by setting stricter regulations concerning AirBnB, by limiting tour bus traffic in residential areas in the capital, and by raising costs for tourists, for example through tourism taxes. The articles also described attempts to distribute tourists better in time and space through, for example, marketing campaigns. The management and improvement of infrastructure, such as sanitary facilities, were also quite frequently mentioned. While there were many articles that specified the ways in which the government and organizations were reacting to tourism, 9.1% of all articles said that they were reacting but did not specify in which ways. In addition, there was one mention that Icelandic stakeholders were not taking action to combat overtourism.

12% of all articles dealt with how tourists have and are able to avoid contributing to the overtourism problem in Iceland. The article writers encouraged tourists to skip traveling to Iceland and travel to other destinations instead, including Greenland, the Faroe Islands, Norway, Finland, or the Japanese Alps. Tourists were also encouraged to travel to less visited parts of Iceland, such as the Westfjords, and to visit the country in the off-peak season. However, there were also three mentions, or around 1% of all articles, that claimed that traveling to Iceland would be a way to escape overtourism in other destinations. These mentions were in the three articles that argued that overtourism was not a problem in Iceland.

Finally, 1% of all articles mentioned how the industry had been responding to overtourism. The articles described how the first five-star hotels were being built as an attempt to cater to luxury

clients and thus contribute to profitability rather than increased tourist arrivals. They also presented interviews with tourism companies who had started to offer small group tours to Iceland with a focus on avoiding the busiest sites at times of peak demand.

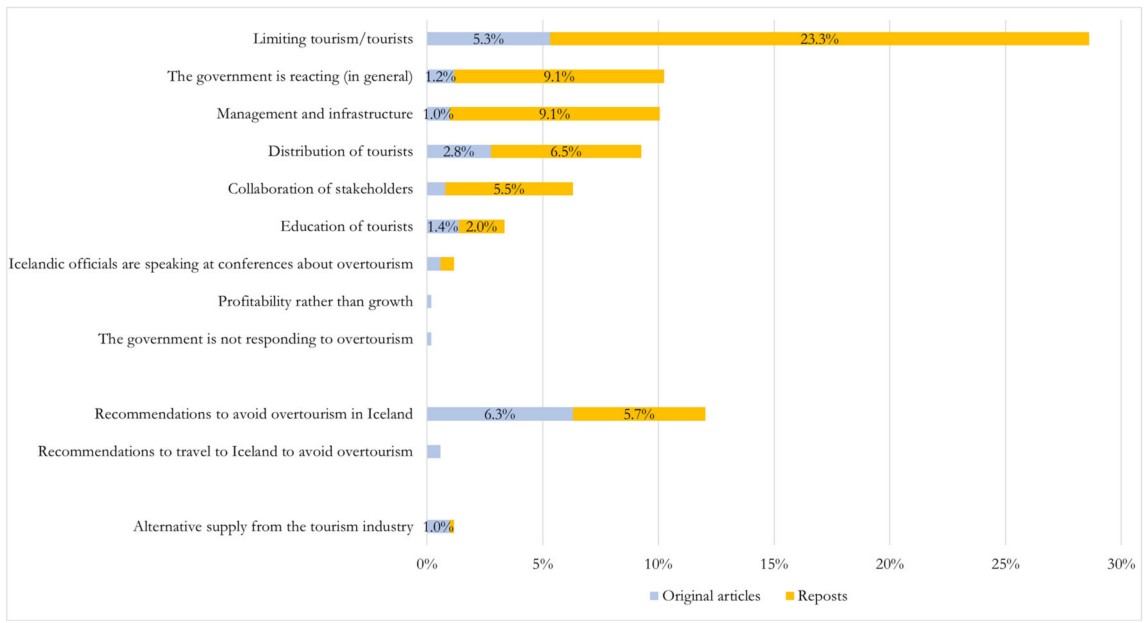

**Figure 7.** Reactions to overtourism (N = 507).

## 6. Discussion

Based on a media content analysis, this paper examines the overtourism narrative that has been constructed around overtourism in Iceland. This is important because the media affects Iceland's identity and image as well as potentially influencing the decision-making of individuals and policy actors. The media has contributed to the social construction of Icelandic nature as pristine wilderness with few people [114] and, as the analysis of media in this paper illustrates, has now also contributed to the idea of Iceland as a destination plagued by overtourism. The extent of media coverage appears to have created a new international discourse of region [57] for Iceland, which represents the country as a notional "poster child" for overtourism [108–110], posits reasons for its manifestation and causes, and is concerned with the management of the supposed overtourism situation.

However, as noted in the paper, Iceland is qualitatively different from many of the other destinations with which overtourism has been connected because it is primarily a nature-based tourism destination, does not have a social movement opposed to tourism, and has a community that is generally supportive of tourism. The media articles that have been analyzed with regard to overtourism generally report on the high growth rate of international arrivals to Iceland from 2010 to 2018 [7,8] and the high ratio of visitors to permanent inhabitants (6.73 tourists per inhabitant in 2018) [16,17], but they only rarely note that residents' attitude towards tourists and the tourism industry is mostly positive [91,94–97]. Similarly, imbalance in reporting is found with respect to the spatial and temporal dimensions of international tourism. Seventy-five percent of articles (75%) in the analyzed media suggest that the country as a whole is a prime example of overtourism, with less than a quarter pointing out that overtourism, in terms of crowding, is more of a seasonal and site-specific problem. Although studies of media reporting of the impacts on tourism are very limited, the results of the present study reinforce the notion that coverage of tourism impacts is often highly selective and sometimes sensationalist, as has been observed with respect to media coverage of tourism and climate change as well as backpacking [115,116].

Several of the themes identified in this study with respect to media representation of overtourism, such as crowding and negative tourist behavior, have also been reported as examples of overtourism in

other media analyses [29,117,118]. However, these other studies primarily provide a generic analysis of articles on overtourism, usually with an urban destination emphasis, rather than focusing on a specific destination [29,117]. As well as having a much larger sample size than other media-related research on overtourism, the present study is therefore the first to look at media representation of overtourism for a particular destination. This is significant as it allows for a more direct comparison of themes in the media versus other information on the state of tourism at the destination.

The above discussion of selective media reporting of certain aspects of tourism in Iceland with respect to visitor growth and the ratio of international tourists to residents therefore highlights the importance of recognizing that what is not reported can be as significant in destination marketing management as what is. In the case of Iceland, the government websites that detail visitor numbers and the economic significance of tourism also report the results of surveys of community attitudes to tourism, but the latter information is rarely conveyed in the media stories that have been studied, perhaps because it is at odds with the writing of a simplistic destination narrative that "fits" with other commentary on overtourism [1,2,29,117]. As visitors' perception of crowding is one of the main manifestations of overtourism [25,29], it is surprising that the analyzed articles fail to discuss visitors' perception of overtourism at some of the most popular nature destinations, even though data on perceptions of Iceland by both residents and tourists are readily available from government websites. Instead, the articles frequently blame social media for Iceland's popularity as a tourist destination, and they especially blame Instagram, with its geotag function. Yet there is little empirical evidence that this is actually the case [21,102]. This observation is not to suggest that there are no significant issues with respect to the large numbers of tourist arrivals in Iceland [102,119]. The Environment Agency of Iceland has closed off a few areas to protect nature due to the pressure of tourism, especially during periods with thawing conditions [120]. Negative tourist behavior, such as human waste and littering, as often reported by Icelandic media over the summer season, also harm the environment [121,122]. Nevertheless, these are issues primarily related to seasonal demand as well as to the area around the capital rather than areas elsewhere in the country, which are actually seeking increased visitation [21].

A feature of this research that has not been reported in any previous analyses of overtourism is the extent of reposting of media stories. This is often the case, for example, with stories produced by specialist news services, such as Reuters, or via intermedia agreements. Although previous research has shown how social media reposting can influence potential tourist generating regions, the life span of travel news, and tourists' attitudes towards travel policy changes [123], there is little research on the reposting and reuse of online media stories (excluding social media) and the reason why some stories are reposted and not others [124]. In the case of the population of media stories examined in the present work, it is noticeable that there is a marked increase in the reposting of stories in October and November, well after the summer season has finished and after the potential overtourism situation that the original story refers to. The extent of news media reposting has not been identified or searched for in previous studies and potentially represents a significant challenge for destination marketing organizations to influence media construction of destination identity and issues [29].

Similarly, while the presence of Icelandic landscapes in popular entertainment, for example in television series, films, and music videos, is considered in some media stories to have increased tourism demand [125,126], there is little actual empirical evidence that this is the case beyond general destination awareness. Several articles mention the television series *Game of Thrones* in particular, as well as the popstar and social media figure Justin Bieber. The increased availability of flights, the advent of low-cost airlines, and the generally lower prices for flights to Iceland are other reasons mentioned as being behind the overtourism phenomenon, while the Icelandair Stopover program and marketing effort is also noted [127–129]. Nevertheless, surveys of international visitors report that it is the broader perceived naturalness of Iceland that draws visitors, including substantial numbers of repeat visitors [21,69,91,102]. Furthermore, surveys also suggest a high level of satisfaction among international visitors with natural areas, with some of the more significant impacts arising because of energy infrastructure rather than tourism developments or tourists [69].

Most of the articles also covered responses to overtourism in Iceland. Most of them dealt with the reaction by the government and official organizations. The coverage was focused, for instance, on stricter regulations concerning Airbnb, limits on bus traffic in residential areas in the capital, and influencing tourist behavior, for example through tourism taxes and marketing campaigns. What is not mentioned in the articles is that, with an increased number of visitors, increased investment has been aimed towards tourism facilities and infrastructure, with more being built than ever before [130]. The response is such that, when international tourism to Iceland eventually picks up again after COVID-19, the country will be in a much better position to manage demand. Indeed, 1% of all articles claimed that traveling to Iceland would be a way to escape overtourism in other destinations or that overtourism was not a problem in Iceland, indicating that representations of overtourism remain relative, while also pointing to a potential future marketing response to influence the media discourse on overtourism that had emerged on tourism in Iceland prior to COVID-19.

## 7. Conclusions

This paper has focused on media discourses on overtourism in Iceland. The results are clearly important in terms of future attempts of the Icelandic tourism marketing bodies to influence the destination discourse on Iceland that had emerged prior to the COVID-19 pandemic, but the results are also significant with respect to gaining a broader understanding of media discourses and representations of overtourism and their implications. Although the future of tourism post-COVID-19 is unknown, it is likely that, in the long-term, overtourism issues will reemerge as a challenge for tourist destinations [20], both in reality as well as in how they are perceived and represented by the media. Indeed, COVID-19 may set new challenges with respect to perceptions of appropriate numbers at a destination or attraction because of the development of new social distancing norms [20]. Therefore, studies such as the one presented here are of high importance in identifying the main representations of an issue such as overtourism in the media and their potential implications. Perhaps significantly, it has identified that a large number of media stories are reposts rather than the original story. As noted, the extent of reposting has not been highlighted in previous media analyses of tourism. Future research therefore needs to engage with media organizations to determine the rationale of reposting specific stories as well as the timing of reposts. It is possible that online media reposting occurs because of the potential of stories to generate views and therefore advertising exposure; if that is the case, it is not so surprising that the coverage of tourism in Iceland is so highly selective as it may generate more interest even if the story is unbalanced [131].

Iceland, like a number of destinations around the world, had become one of the examples of overtourism used in the media. However, Iceland is qualitatively different to other examples of overtourism, such as Barcelona and Venice [2], as there was never organized opposition and social movements against tourism. Indeed, as noted, while Icelanders were concerned about the effects of tourism, surveys showed they remained in favor of continued international tourism [94–97], something that was not readily picked up in media stories of overtourism in Iceland even though the information is readily available from government websites that contain other information used in online media stories. The overtourism media discourse surrounding Iceland was therefore a very partial picture of the on-the-ground reality that failed to take into consideration the spatial and temporal nature of the concentration of tourists. Many parts of the country were consequently unaffected by overtourism issues for most of the year and, in fact, were seeking to encourage more tourists, which could even be described as undertourism. In addition, the provision of improved visitor infrastructure and facilities received little media attention, while even though a minority of tourists did report crowding at some locations, they nevertheless continued to report that their experience in Iceland was extremely positive and that they perceived Iceland as a destination with strong natural and wilderness features. There are therefore substantial discontinuities between media representations and discourses of tourism on Iceland and local discourses and understandings, a finding that reinforces the results of other media analyses of overtourism [29,117] and that potentially also sees the need for a more

focused response in destination marketing and management terms. However, unlike previous content analysis of media coverage of overtourism [29], the present study focused on stories in relation to a specific destination and provided a more detailed analysis of destination-specific themes as opposed to more general concern regarding tourism growth [1,2,29]. As such, the present study contrasted media stories surrounding a destination with other destination-specific data, indicating a significant divide between destination narratives and representations.

Prior to the advent of COVID-19, media representation of overtourism and Iceland was being blamed for a decline in international visitor arrivals [132], although this was seemingly being done in the absence of specific empirical research. Future research is required to understand how the overtourism discourse becomes reinterpreted over time, especially as media coverage and policy response tends to move through issue-attention cycles [20]. Representations, although not reflecting the reality of some tourism stakeholders or the results of empirical research, still matter. Future analysis of media stories would greatly benefit from more detailed examination of the diffusion of media stories and identifying the relative importance of the original sources of stories in reposts. From a media and marketing influence perspective, not all media is equal. Influence is not based just on the immediate number of readers but also on how many times a story is repeated in other media. Therefore, destination marketing organizations and others who seek to influence media agenda-setting and cycles may need to reassess media communication strategies in order to both influence and respond to particular destination representations [133]. Indeed, a limitation of the present study with respect to issues of agenda-setting and issue-agenda cycles is that it only examines one destination over a one-year period. Ideally, future research should take a multidestination approach and look to identify narrative change and destination representation in the media over time.

Greater work is needed on connecting media discourses to actual consumer decision-making, while a clear issue is the extent to which stories become republished and reused until they almost become self-fulfilling, i.e., if something is mentioned in the media often enough, it then becomes "true" as a result of narrative repetition, although whether this is a case of deliberate manufacture or not remains a moot point [134–136]. Nevertheless, the traditional journalistic focus on novelty has broken down in recent years, given increased reuse of stories for economic reasons and the advent of social media [137,138]. Related to this is the importance of finding out from the writers and publishers why they decided to focus on particular aspects of the overtourism story and not others. However, an additional observation here is that it is not only the media but also academics that reinforce the "relevance" or "utility" of the concept of overtourism. The term, which, as noted, was originally invented by a marketing and public relations firm, described long-identified issues in tourism development [23]. In the same way that terminology or issues become fashionable in the media, so this is also the case in academic research, where fashion cycles and bandwagon effects may influence research interests and the framing of research [139]. By increased reference to overtourism in their publications and reports, researchers, like news media, may unwittingly strengthen the relevance of an idea even though its actual application may otherwise be highly restricted in both space and time.

This study therefore reinforces the notion that overtourism, along with the common pool resources that are often the focus of concerns over there being too much tourism, reveals the importance of the need to better understand the different and often contested discourses that surround destinations, as discourses and media representations of them can have real effects on both tourist perceptions of place and, therefore, their decisions to travel as well as on how destinations choose to manage and market tourism.

**Author Contributions:** Writing, review, and editing, A.D.S., C.M.H., and M.W. All authors have read and agreed to the published version of the manuscript.

**Funding:** This research received no external funding.

**Acknowledgments:** We thank Sigríður Dögg Guðmundsdóttir at Promote Iceland for the assistance with the online search with the program Cision.

**Conflicts of Interest:** The authors declare no conflict of interest.

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
