# Peer review of "Overtourism in Iceland: Fantasy or Reality?"

_sustainability, doi:10.3390/su12187375_

Round 1

Reviewer 1 Report

The topic of this paper is relevant to this sustainability journal. The researchers explore over-tourism in Iceland. Please find enclosed my constructive remarks for your guidance.

From the outset, I noticed that the researchers need to submit their manuscript to a professional proofreader. It has grammatical mistakes.

The abstract should better accentuate the implications of this research.

The introduction: The researchers have provided a decent background to their paper on over-tourism. However, I would suggest that they make more reference to sustainable tourism (or responsible tourism). I would strongly recommend that they elaborate further on the researcher question of this paper. They need to build on the relevant literature. Yet, they should explain how this contribution addresses a gap in academia.

The literature review can include more papers on sustainable tourism. I suggest that they feature the following as well:

Miller, G., Rathouse, K., Scarles, C., Holmes, K., & Tribe, J. (2010). Public understanding of sustainable tourism. Annals of tourism research37(3), 627-645.

Camilleri, M. A. (2016). Responsible tourism that creates shared value among stakeholders. Tourism Planning & Development13(2), 219-235.

Boley, B. B., McGehee, N. G., & Hammett, A. T. (2017). Importance-performance analysis (IPA) of sustainable tourism initiatives: The resident perspective. Tourism Management58, 66-77.

The authors should provide more details about the rigour and trustability of their research in terms of their data capture and analysis (to ensure that their study is adaptable and can be replicated in other contexts). E.g. They reported that "the qualitative analysis assessed the meaning behind themes and patterns". What were the themes and the patterns? They need to clarify their statements. What about CISION? Why did they choose it? Is it reliable?

The results of this study as well as the discussion sections were purely descriptive. The researcher were expected to be more argumentative and should have provided a better interpretation of the findings. To be fair in they featured a discussion in their conclusions. However, in this case, they have merely reiterated previous discourse on over-tourism.

I suggest that the researchers include a small section to discuss about the limitations of their paper. They can also identify future research avenues.

Reviewer 2 Report

I am impressed by the authors’ manner of writing this paper: it is a very fluent and coherent manuscript, which uses a perfect English. Also, the structure and flow of the paper make it easy to read, while readers can fully understand authors’ ideas.

Despite its logical structure, fluency and coherence, the paper has some flaws which authors should address. These flaws are related to the paper’s theoretical contribution/relevance and its practical implications.

The paper addresses a very interesting and relevant topic. These days many would say overtourism is history and not a relevant topic anymore, due to the COVID-19 pandemic context. Personally, I consider that analyzing the tourism situation on medium or long term, we should acknowledge that overtourism issues will continue to come back within tourism destinations. Authors should somehow point out in the paper the importance of overtourism for future tourism development. They should explain why studying overtourism related issues is (still) relevant for the future of tourism.

In what concerns the paper’s theoretical contribution authors state in their conclusions section that their results are significant with respect to gaining a broader understanding of media discourses on overtourism and their implications. This is simply not enough to prove the paper’s scientific relevance. How does the paper/the research contribute to extant literature? Does it bring anything new to existing theories or knowledge of the subject? (not specifically to Iceland). Authors say that other researchers previously conducted similar research, based on content analysis on news articles, with the aim of shedding light on the media frames overtourism. Then, what do they do differently? Just focusing on another tourism destination does not contribute to the literature significantly. All these issues should be addressed in the paper, preferably in the introduction section, and then reiterated with a bit more details in the discussion section.

Considering its practical implications, even though I can say that the paper has potential in this direction, authors should better state their case. At the beginning of the paper, authors say that destination discourses can play an important part in influencing tourist decision-making and government and industry policy making. In their conclusions section they assert that results are important in terms of future attempts of the Icelandic tourism marketing bodies to influence the destination discourse on Iceland. However, that’s pretty much it. Therefore, authors should further argue and develop their ideas regarding the practical implications of their study. How would the study help Icelandic tourism marketing bodies to influence the destination discourse on Iceland? Authors should explain, extend and argue specifics in this direction. They should also try to extrapolate, if possible, these practical implications to other tourism destinations worldwide.

Reviewer 3 Report

Dear authors

Thank you for the opportunity to read your interesting and timely paper. The idea to progress the understanding of overtourism by media analysis is of special importance, as media are an important subject  in the overtourism debate and communication. The approach is well suited, by content analyses that explore overtourism narrative in Iceland. In this respect, the first part of the purpose of the paper (to  analyse the media discourse on overtourism in Iceland) has been conducted  very well. However, the presentation of the part two of the purpose (how overtourism reveals itself based on existing studies on crowding and carrying capacity in Iceland) has not been well structured. I found many new information from previous studies in the Discussion section – and I would expect them to be presented in your state of the art sections or in a section on Study area – in order to set the informed background for your discussion. At the same time, your Background section on overtourism state of the art is somehow quite broad and not well focused on the presentation, needed for your study purpose. For example, why are the norms presented in much details, why are they needed for this study? And if so, I think that SET (social exchange theory) or representation theory is also important (and not presented). I am not sure that norms can be used to define the number of people a person can tolerate. I think it is a socio-psychological capacity that needs social indicators to be measured (see studies on quality of life or wellbeing, example  by  Moscardo, G. (2009), “Tourism and quality of life: Towards a more critical approach”, Tourism and Hospitality Research, Vol. 9 No. 2, pp. 159–170,  for example).

In total, improvements in Background section are possible, if you accept my argument and decide to straighten and focus the provided state of the art. However, improvements in discussion section are recommended. Overall the paper is well written, and the study adds to our knowledge body and understanding of overtourism phenomena in the case of Iceland. Thus, I suggest minor changes and strongly support the publication of this paper in the Sustainability.

Reviewer 4 Report

Dear authors:

Thank you for sending us your proposed article entitled "Overtourism in Iceland: fantasy or rality?"

At first glance, the chosen topic is interesting and cool, in this moment when the debate on overturism is one of the issues that mark the tourist and economic agenda of many territories.

As a first point, another type of title should be proposed that makes a more specific reference to the work carried out. It is not very academic and choosing an appropriate title can help in the searches that the experts carry out on the subject.

As for the introductory chapter, I believe that a general framework on the subject of study should be provided, as well as a description of the physical and economic geography of Iceland, providing relevant data on the tourist phenomenon in historical series. A situation map of Iceland with its main cities and natural parks (mentioned in the work) would be a very interesting graphic support to really know the dimension of the problem and to be able to locate the places mentioned in the text.The introduction directly addresses the concept of overtourism, a question that due to its importance and dimension should be included in a specific section on the state of the art, where the basic concepts on which the paper revolves are established. In the text, it is pointed out that overtourism is a concept that begins to be discussed in 2016, however this concept can be traced in the scientific literature much earlier with important and interesting contributions.

In this review of concepts, a series of sections should be established that also address the concept of load capacity, mentioning contributions and measurement methodology, as well as saturation. The contributions that are made are erratic and do not follow a coherent discourse in time or in the contributions, although it is true that relevant studies are mentioned.

In the paragraph comprised by lines 62 to 71 there are a series of questions that I understand are intended to be answered with the research, so they should be clearly outlined as research questions.

In the same way, a broad vision should be established on similar works that analyze the influence of the media on the perception that a destination may be "overtourimed"

If we address the concept of carrying capacity, the carrying capacity of the various tourist attractions in Iceland should be established. The measurement of the number of tourists can be very misleading, depending on the places they choose to visit, stay ...

As for point 3, which refers to the study area, as I have indicated it should be a section of the introduction providing information of interest. Likewise, graphs should be implemented in historical series for a better understanding of certain concepts (income per capita from tourism), number of tourists, hotel capacity, number of overnight stays, number of days spent overnight in destination, relationship between tourist income and GNP , among others. Figure 3, due to its generality, is not very illustrative and poor.

Regarding the methodology, it is mentioned that the research is carried out through content analysis. This technique should be better explained by mentioning relevant authors (Mass Media Research: An Introduction Roger D. Wimmer, Joseph R. Dominick, 1996) and the limitations of this method.

When using a content analysis methodology, the description of the selected media and their characteristics is very important. It is very useful and interesting to establish an analysis matrix that includes some of the following data: Type of Media (Social Media, Written, Radio, Television ...), Audience, Scope. Likewise, it is interesting to know the graphic content of the published news, to see if they really illustrate the concepts studied.

An important limitation of the study is that the frequency of use of the terms analyzed is indicated, but since the audience of the medium is unknown, the real and direct impact of said publication is unknown. Likewise, an attempt should be made to observe the rebound of information that results in its replication on social media, which has an amplifying effect.

Using the proposed methodological design, the results are correct, but due to the proposed methodological poverty they could have been much more interesting and illustrative.

As for the discussion, results, lines 470-499 and some conclusions are mixed in it, in a place where divergent questions with similar studies described in the state of the art should be established.

The concussions should be more specific, answer and extend to the research questions.

In the margin I would like to point out that the magazine where the paper is to be published is called "Sustainability", a concept that appears rarely in the text. One of the questions that should have been raised is whether the Icelandic tourism model is sustainable and what relationship the media establishes between overtourism and sustainability, a question that opens an interesting debate in the Discussion and in the conclusions.

Reviewer 5 Report

Thank you for the opportunity to read this interesting submission on overtourism in Iceland. The paper is extremely well written and interesting. I only have a few minor comments for improvement. Firstly, the paper (although generally well written) has a few typos and minor errors in English grammar and expression. The literature review is quite comprehensive, but agenda setting, framing and the issue attention cycle are only referred to in the conclusion and should probably be discussed in this section first. The background section on Iceland (3 Study Area) is perhaps a little too long and detailed and as the paper is quite long anyway, this section could be condensed. In the Methods section, the discussion of how the data were analysed is a little short - for example, if the full content of each article was the data point (including the headings and the photos) how were the photos content-analysed? The Discussion section is rather repetitive of the findings, and needs to be improved by incorporating references back to the existing literature. In the Conclusion section it would be good to see more discussion of the implications of this work outside the specific context of Iceland - there is some mention of the tragedy of the commons in the conclusion, but this could be expanded to assist with the generalisability of the results to other overtourism contexts.

Round 2

Reviewer 1 Report

The researchers have revised their manuscript in different areas. However, these changes are not adequate and sufficient.

The results of this study as well as the discussion sections are still not rigorous enough. This is a descriptive paper. It cannot be published in this high impact journal. I would suggest that the researchers adopt better research methods to  ensure that their study can be replicated in other contexts. 

Quantitative studies are evaluated in terms of their validity and reliabliiity of their constructs. Whilst qualitative studies ought to be rigorous and trustworthy, terms of their dependability, confirmability, transferability and adaptability. 

In your case, your manuscript is purely descriptive. Notwithstanding, the methodology and the results are very, very weak.

For this reason, I cannot recommend the publication of this paper. I would reject it from this high-impact journal.

Reviewer 4 Report

Dear authors, thank you for including the suggestions made. I really appreciate the tables that have been added that help to better understand the work. In the same way, now the state of art is richer and offers a complete vision of the problem addressed. I am sure that your research will be of interest to the scientific community.

Round 3

Reviewer 1 Report

I noticed that you have refined your descriptive paper. You have taken heed of the reviewers' constructive remarks. Hence, you have improved it in many areas. The paper is publishable now.

The final decision rests with the editor.